# Identification and Functional Analysis of LncRNAs in Response to Seed Aging in *Metasequoia glyptostroboides* by Third Generation Sequencing Technology

Yongjian Luo [1,2,3], Jingyu Le [1,3], Yixin Zhang [2], Ru Wang [1,3], Qing Li [2], Xinxiong Lu [4], Jun Liu [2,*] and Zhijun Deng [1,3,4,*]

1   Hubei Key Laboratory of Biologic Resources Protection and Utilization, Hubei Minzu University, Enshi 445000, China
2   Agro-Biological Gene Research Center, Guangdong Academy of Agricultural Sciences, Guangzhou 510640, China
3   Research Center for Germplasm Engineering of Characteristic Plant Resources in Enshi Prefecture, Hubei Minzu University, Enshi 445000, China
4   Center for Crop Germplasm Resources, Institute of Crop Sciences, Chinese Academy of Agricultural Sciences, Beijing 100081, China
*   Correspondence: liujun@gdaas.cn (J.L.); dengzhijun@hbmzu.edu.cn (Z.D.)

**Abstract:** The seeds of *Metasequia glyptostroboides* Hu et Cheng, an endangered species, are susceptible to aging, making natural population renewal difficult and increasing the risk of extinction. LncRNAs play important roles in plant growth and development and biotic and abiotic stress responses, but the functions of lncRNAs in the aging process of *M. glyptostroboides* seeds are still unclear. In this study, we used single molecule real-time (SMRT) sequencing technology in combination with Illumina RNA-seq to analyze lncRNA changes during *M. glyptostroboides* seed aging. We identified 403 intergenic lncRNAs (lincRNAs), 29 intronic lncRNAs, and 25 antisense lncRNAs; screened 9000 differentially expressed mRNAs (DEGs) and 128 differentially expressed lncRNAs (DELs); and predicted 844 cis-target genes and 8098 trans-target genes. GO and KEGG functional annotation of target genes revealed that the regulation of the reactive oxygen species metabolic process, protein processing in the endoplasmic reticulum, and the MAPK signaling pathway and other pathways were significantly enriched, showing a high correlation with the mRNA enrichment results. In addition, we constructed a ceRNA network consisting of 18 lncRNAs, 38 miRNAs, and 69 mRNAs, in which some miRNAs and mRNAs related to seed aging were found. Among them, miR167(a,b,c,d) may compete with lncRNA_00185, which is related to plant aging, to regulate the expression of the *RCD1*(Radical-induced Cell Death1) gene, thus promoting the balance of seed reactive oxygen species and enhancing seed-aging resistance. These results will have significant reference value in elucidating the molecular mechanism of the seed aging of *M. glyptostroboides* sequoia, improving the storage capacity for crop seeds, and protecting rare germplasm resources.

**Keywords:** seed aging; *Metasequia glyptostroboides* Hu et Cheng; long non-coding RNA (lncRNAs); ceRNA; SMRT

## 1. Introduction

Seeds are the foundation and key to agricultural production, and the quality of seeds plays a vital role in agricultural and animal husbandry production, the effective utilization of economic and genetic resources, biodiversity conservation, and the restoration and reconstruction of plant communities. At the peak of physiological maturity, seeds must undergo a comprehensive process of inevitable and irreversible changes in vigor, namely seed deterioration or aging [1,2]. Studies have shown that a series of detrimental changes occur internally during seed aging, such as decreased seed-coat mechanical tolerance [3],

oxidative damage to cell membranes, degradation of soluble sugars and proteins, DNA damage and mutation, and disruption of nucleic acid synthesis systems [4–7], eventually leading to the loss of seed vigor. Seed aging is not only related to seed and seedling growth, yield, and quality, but also has a serious impact on the conservation, utilization, and development of germplasm resources, as well as agricultural production [1,2].

*Metasequoia glyptostroboides* Hu et Cheng, as the only extant relic species of the *M. glyptostroboides* genus in the family Taxodiaceae, is an endangered endemic tree species under first-class protection in China, and has been successfully introduced into more than 50 countries and regions [8]. However, 98% of the existing relict species are distributed in the Xiaohe Valley area of Lichuan City, Hubei Province, China, and the age structure is in an inverted pyramid shape [9] and a random distribution pattern, which makes it difficult for *M. glyptostroboides* populations to alternate between new and old in their natural state [10]. Based on this, Liu et al. speculated that the low seed vigor and poor vigor retention ability of *M. glyptostroboides* may be one of the important reasons why natural populations are difficult to regenerate [11]. It was found that the average germination percentage of newly harvested *M. glyptostroboides* seeds was only (32.9 ± 3.3)% [12], the average seed viability was low, and under artificially accelerated aging conditions, the loss of *M. glyptostroboides* seeds was extremely rapid [11]. With global warming, there will be more days of high temperature and high humidity that will accelerate the aging of seeds, reduce their vigor [7], severely limit their natural regeneration ability, and greatly increase their extinction risk [12,13]. At present, there are many related studies on seed vigor, but most of them focus on the seeds of major crops such as corn, soybean, tomato, and rice. There are few studies on the vigor of forest tree seeds, especially for the changes of seed vigor of wild plants under natural conditions. The aging law and its vigor loss mechanism are even less studied [14,15]. Therefore, it is of great significance for the protection of precious germplasm resources and the improvement of crop-seed storage capacity to clarify the aging law of *M. glyptostroboides* seeds, to mine the genes involved in seed-aging resistance, and to clarify the molecular mechanism of seed-aging adaptation.

In recent years, with the rapid development of high-throughput sequencing technology, thousands of aging-related lncRNAs have been discovered [16,17]. LncRNAs are a class of RNAs with a length greater than 200 nt and which basically do not have the ability to encode proteins. LncRNAs can regulate gene expression at different levels through cis- and trans-acting (cis/trans), e.g., they can manipulate nuclear domains [18], post-transcriptional modifications [19], variable splicing [20], transcriptional interference [21], protein modification, DNA methylation, and so on. Studies have confirmed that lncRNAs play an important role in plant growth and development, and in response to biotic and abiotic stresses [22]. For example, *lncRNA SABC1* can inhibit the expression of transcription factor *NAC3,* control the biosynthesis of salicylic acid (SA), and balance plant defense and growth [23]. In the process of rice-endosperm nuclear division and endosperm cellularization, *lncRNA MISSEN* can hijack a helicase family protein *HeFP* to regulate the function of tubulin, resulting in abnormal cytoskeleton polymerization, which in turn affects early rice-endosperm development [24]. *LncRNA ELENA1* could participate in the immune response of Arabidopsis thaliana [25]. Ye et al. found that the expression of *Ptlinc-NAC72* was strongly up-regulated in the late stage of salt stress, and believed that it could be important for plant growth recovery after salt stress by a preliminary study on its function in poplar [26]. LncRNA *MSTRG.19915* can form complementary double strands with the transcript of *BrMAPK15*, thereby regulating the transcription level of *BrMAPK15*, and then participating in the disease-resistant immune response of cabbage downy mildew [27]. At present, although the functions and characteristics of lncRNAs in some model crops have been identified, due to the low sequence conservation of lncRNAs and high tissue specificity, the potential functions of lncRNAs in plant seeds are still unclear [28], and the mechanism by which lncRNAs respond to seed-aging stress remains to be studied.

Despite rapid advances in sequencing technology, the whole genome assembly and sequencing of large genome plants such as *Sequoia sempervirens*, *M. glyptostroboides*, and

*S. giganteum* is still a huge challenge [29]. The lack of high-quality genomic maps greatly limits research into plants with large genomes [30]. The use of SMRT sequencing technology can provide ultralong reference sequences for RNA-Seq studies without a reference genome, providing a new possibility for the research of some plants [31]. In this study, we used the SMRT sequencing technology of the PacBio platform to perform full-length transcriptome sequencing, and combined with illumina high-throughput short-read sequencing technology to obtain complete full-length transcripts and lncRNAs, to comprehensively analyze the function of target genes of differentially expressed mRNAs and lncRNAs and the relationship between miRNAs, lncRNAs, and mRNAs, so as to explore the lncRNA changes in the aging process of *M. glyptostroboides* seeds, and provide new ideas for solving the mystery of the seed-senescence mechanism. Understanding the mechanism of *M. glyptostroboides* in response to senescence stress can also provide a theoretical basis for the further protection of endangered plants and ecological restoration.

## 2. Materials and Methods

### 2.1. Plant Materials and Ageing Treatments

Fresh mature *M. glyptostroboides* seeds and plant samples were collected from Xiaohe, Lichuan, Hubei (N: 30.300042, E: 109.509973) after frost in 2021, and were artificially aged for 4 days under conditions of high temperature and high humidity (40 °C, 100% relative humidity) [32]. Aged *M. glyptostroboides* seeds (S4) and untreated fresh seeds used as the control group (S0) were snap-frozen with liquid nitrogen and stored at −80 °C for subsequent experimental assays. Each treatment was repeated three times.

### 2.2. Extraction of Total RNA

The total RNA of S4 and S0 samples and full-length transcriptome samples (roots, stems, leaves, flowers, seeds) were extracted using a Tsingke RNA kit, and the contamination and degradation of RNA samples were detected by agarose gel electrophoresis. RNA concentrations were measured using a NanoDrop 2000 spectrophotometer (Thermo Fisher Science, Wuhan, Hubei, China). RNA integrity was assessed using the Agilent 2100 system (Agilent Technologies, China). The total RNA purity, i.e., the value of OD 260 nm/OD 280 nm, was tested to be in the range of 1.8–2, the value of OD 260 nm/ OD230 nm was greater than 0.9, and the RNA integrity was greater than or equal to 7.5, meeting the requirements of transcriptome sequencing.

### 2.3. PacBio SMRTbell Library Preparation

In order to obtain the full-length transcriptome sequence of *M. glyptostroboides*, the total RNA from the roots, stems, leaves, flowers, and seeds of *M. glyptostroboides* vulgaris was completely mixed in equal amounts to construct a sequencing library. RNA was first reverse transcribed into cDNA using the SMARTER™ PCR cDNA Synthesis Kit. After 14 cycles of PCR amplification, library construction was performed using the BluePippin™ Size Selection System (Sage Science, MA, USA) to screen for 1–4 kb and >4 kb products. The library was then subjected to large-scale PCR amplification of full-length cDNA for SMRTbell library construction (including cDNA damage repair, end repair, and ligation with SMRT dumbbell adapters). Before sequencing, the template was annealed with sequencing primers, the polymerase was ligated to the template after primer annealing, and sequencing was performed on the PacBio Sequel platform.

The raw data were initially processed using the Iso-seq standard pipeline technique using SMRTlink (v7.0) to obtain subreads. Circular consensus sequence (CCS) fragments were extracted from subreads in the BAM file of the off-camera data. Secondly, according to whether the 5′ primer/3′ primer/polyA was complete and whether it was chimeric, the CCS was divided into a full-length non-chimeric (FLNC) sequence, full-length chimeric (FLC) sequence, non-full-length (NFL) sequence, etc. LoRDEC software was used to read the Illumina second-generation short reads data of six samples to correct the full-length transcriptome, and CD-HIT software was used to remove redundant sequences

through sequence alignment and clustering to obtain non-redundant transcript sequences for subsequent analysis.

### 2.4. Functional Annotation and LncRNA Identification

To investigate the functions of all the non-redundant transcripts, BLAST (v2.2.26), KOBAS (v3.0.0) and HMMER (v3.3.2) software tools were used to search the following public databases: Non-redundant protein sequences (NR), Nucleotide Sequence Database (Nt), EuKaryotic Orthologous Groups (KOG), Cluster of Orthologous Group (COG), Protein Sequence Database (Swiss-Prot), Kyoto Encyclopedia of Genes and Genomes (KEGG), Gene Ontology (GO) and protein domain database (Pfam). To identify lncRNAs in the Iso-seq data, we used four analysis methods, namely Coding Potential Calculator (CPC), Coding Non-Coding Index (CNCI), the Protein Families database (Pfam) and Predictor for long non-coding RNAs and messenger RNAs based on an improved k-scheme (plek), which was used to assess the degree and quality of the open reading frame (ORF) in transcripts. The E value was set to "$1 \times 10^{-10}$", and NCBI eukaryote-protein database was used to search sequence to distinguish between coding and non-coding transcripts. CNCI was set to default parameters. The setting parameters for searching Pfam were "$1 \times 10^{-5}$". PLEK used the default parameter of minlength 200 to evaluate the coding potential of transcripts that lack genome sequences and annotations, and deleted transcripts with a predicted length of <200 bp.

Transcripts without coding potential were selected as our candidate lncRNAs. Due to the lack of a complete genome sequence of *M. glyptostroboides*, lncRNAs sequences were aligned to *Sequoiadendron giganteum* and were classified into four categories, including sense lncRNAs, lincRNAs, antisense lncRNAs, and sense-intronic lncRNAs.

### 2.5. Data Analysis

Gene expression levels were identified by RSEM v1.3.0 for six samples. LoRDEC software was used to map RNA-seq data to Iso-seq data to obtain the corrected consensus sequences. To determine the gene expression levels in a given response to seed aging, the corrected consensus sequence was de-redundified using CD-HIT ($-c\,0.95 -T\,6 -G\,0 - aL\,0.00 -aS\,0.99$), and the ensuing full-length transcripts used as a ref for that particular gene. Subsequently, the clean reads obtained by Illumina sequencing were aligned to reference, and the read-count of all genes was obtained. Genes with fragments-per-kilobase of transcript-per-million mapped read (FPKM) values >0.3 in samples from two groups (aging stress and control) were selected for further analysis. The DESeq2 R (v1.32.0) software package was used for the expression analysis. In order to control the false discovery rate, Hochberg and Benjamini were used, and the DEGs were designated as those having FC of $|\log_2 FC| > 1$ and *p*-value < 0.05.

### 2.6. Predicting the Potential Target Genes of lncRNAs and Functional Enrichment Analysis

The prediction of cis-target genes was based on the principle of sequence complementary pairing; we used blast alignment to obtain mRNAs complementary to lncRNAs (setting E-value = $1 \times 10^{-5}$ and identity = 80%) and then calculated the thermodynamic parameter values after complementary pairing of lncRNAs and mRNAs using RNAplex software, and selected the result above the software-threshold range as the cis-target genes of lncRNAs. The correlation analysis of trans-regulated genes was performed by using the FPKM of lncRNAs and mRNAs, and those with a Pearson correlation coefficient (PCC)greater than 0.95 and *p* value less than 0.001 were considered trans-regulated target genes, which were visualized using Gephi (v0.9.7) software.

In order to predict the function of lncRNAs in response to aging stress, gene ontology (GO) enrichment analysis of co-located or co-expressed protein-coding genes of DELs were implemented by the GOseq R package (v1.46.0), and GO terms with a *p*-adj value less than 0.05 were considered significantly enriched. Kyoto Encyclopedia of Genes and Genomes (KEGG) pathway enrichment analysis was conducted using KOBAS software.

### 2.7. CeRNA Network Construction

The mature miRNA sequences were downloaded from miRBase (http://www.mirbase.org/, (accessed on 15 February 2022)). The lncRNAs and mRNAs in the co-expression network (person >0.95, *p* < 0.001) were then used as the ceRNA network miRNA prediction library in lncRNAand mRNA. RNAhybrid (https://bio.tools/RNAhybrid; accessed on 15 September 2022) software was used to predict the target prediction of miRNA in lncRNA, and the program parameters referred to Song et al. [33]. The prediction of the target miRNA of mRNA adopted the online psRNATarget (https://www.zhaolab.org/psRNATarget/analysis; accessed on 15 September 2022); for the specific parameters please refer to Song et al. [33]. The ceRNA (lncRNA-miRNA-mRNA) regulatory network was visualized using Cytoscape (v3.7.2) software.

### 2.8. qRT-PCR Analysis

To verify the accuracy of RNA-SEQ sequencing results, six DELs were selected and Primer5 (V5.5.0) software was used to design primers (Table S1). Total RNA was obtained by a plant TOTAL RNA Extraction Kit (Tiangen, Wuhan, China), and cDNA was synthesized by a reverse transcription kit (Tiangen, Wuhan, China). PCR amplification was performed by rotor-Gene3000 real-time PCR system (Agilent, CA, USA) for quantitative real-time PCR (qPCR), and three biological replicates were set for each experiment. The reaction system was carried out according to the instructions of SYBR®PremixExTaq™II Kit (EnzyArtisan, Wuhan, China). With *M. glyptostroboides* actin as the reference gene, the amplification results were calculated by the $2^{-\Delta\Delta Ct}$ method.

## 3. Results

### 3.1. High-Quality Full-Length Sequencing Provides Sufficient Sequences for Further Analysis

To reveal the genes and regulatory LncRNAs that responds to seed aging in *M. glyptostroboides*, we sequenced and analyzed 4-day-aged (germination percentage 35.42 ± 2.4%) *M. glyptostroboides* seeds and control seeds (germination percentage 58.75 ± 2.4%) using the PacBio sequencing platform. A total of 11,487,282 (32.18 Gb data) sub-reads were generated by SMRT, with an average read length of 2802 nt and N50 of 3108 nt. To avoid loading bias, we obtained 330,124 cyclic consensus sequences (CCS) and average read length from reads with at least two full-pass inserts of 2490 bp (Table 1). A total of 292,684 full-length sequences and 289,659 full-length non-chimeric (FLNC) reads were detected by SMRTlink (v7.0) software with an average of read length of 2314 bp. The similar FLNC reads were clustered using the ICE algorithm. The consensus sequence was obtained, the non-full-length sequence was corrected by arrow software to obtain the consensus sequence, and finally 77,558 polished consensus sequences were obtained, with a mean length of 2339 bp (Table 1).

The RNA-seq experiments generated approximately 119,930,400 and 125,220,590 raw sequence reads from samples S0 and S4, respectively, with 117,531,792 (S0) and 122,716,178 (S4) clean reads after trimming. The raw reads of Illumina sequencing data were then used to correct the SMRT data (Table S2). After removing redundant and similar sequences, we obtained a total of 217,281,041 nucleotides and 42,189 transcripts with a mean length of 2592 bp (Table 1). Overall, 40,446 transcripts (corrected isoforms) were subjected to functional annotation by searching the Non-Redundant Protein Database (NR), Nucleotide Sequences (Nt), Swiss-Prot, the GO, Cluster of Orthologous Groups of proteins (COG), Cluster of Eukaryotic Orthologous Groups (KOG), and the Protein Family (Pfam) and KEGG databases, from which a total of 40,446 transcripts (95.87%) were successfully annotated (Figure S1, Table S3). We also analyzed homologous plant species by comparing the transcript sequences to the ones in the NR database, and found that the majority of transcripts were expressed in *Picea sitchensis* (Figure S2). Lastly, lllumina sequencing data were remapped with the SMRT data. The resulting full-length transcripts were used as a reference, and all valid reads were obtained from mapping short-read sequences to refer-

ence using RSEM software. After mapping, we converted the read counts into FPKM to measure the transcript level for each unigene.

**Table 1.** Summary of reads from third generation long-read sequencing.

| Item | Number |
|---|---|
| Subreads base (G) | 32.18 |
| Subreads number | 11,487,282 |
| Average subreads length | 2802 |
| N50(subreads) | 3108 |
| CCS | 330,124 |
| Number of 5′ primer reads | 309,819 |
| Number of 3′ primer reads | 309,759 |
| Number of poly-A reads | 375,101 |
| Number of non-full-length reads | 34,773 |
| Number of full-length reads | 295,351 |
| Number of full-length non-chimeric reads | 289,659 |
| Average length of Flnc reads (bp) | 3048 |
| Number of consensus reads | 77,558 |
| Average length of consensus reads (bp) | 2799 |
| Total nucleotides (bp) before and after correction | 217,011,161/217,281,041 |
| Mean length before and after correction (bp) | 2799/2802 |
| Minimum length before and after correction (bp) | 68/68 |
| Maximum length before and after correction (bp) | 14,493/14,555 |
| N50 length before and after correction (bp) | 3109/3114 |
| Number of unigenes | 42,189 |
| Average length of unigenes | 2814 |

### 3.2. Identification and Recognition of LncRNAs

Transcript-coding ability is assessed using the coding-potential calculator (CPC), coding-non-coding index (CNCI), coding-potential assessment tool (CPAT), and protein-coding domains in the Pfam database (Pfam) software, and 457 lncRNAs were screened to identify the types of candidate lncRNAs (Figure 1A). As a result, 403 intergenic lncRNAs (lincRNAs), 25 antisense lncRNAs, and 29 introinc lncRNA were found by aligning with the genome sequences of the closely related species *S. giganteum* (Figure 1B). We explored the expression patterns of *M. glyptostroboides* seeds in the S4 and S0 group. In the S0 group, the expression of lncRNA was lower than that of mRNA, while the expression of lncRNA and mRNA were similar and up-regulated in seeds under aging stress (Figure 1C). We then analyzed the sequence characteristics of lncRNAs and mRNAs, and found that the length of lncRNAs was between 252 and 6056 nt, and the average length was 2057 nt. The length of mRNA sequences ranged from 150 to 9537, with an average length of 1536nt. A total of 74.60% of mRNAs genes ranged from 0 to 2000, which was shorter than lncRNAs. Among the ORF lengths, the ORF lengths of lncRNAs are between 0 and 531, with an average length of 200 nt, and the number of genes is between 150 and 200 (Figure 1D). The ORF lengths of mRNAs are between 0 and 9537, with an average length of 1488 nt; the maximum number of genes is between 800 and 1000 (Figure 1E). Correlation analysis of the six samples indicated that the correlation value ($R^2$) of three replicates in the S0 was greater than 0.959, whereas the value for the aging-treatment group was greater than 0.875 (Figure 1F), suggesting that the repeatability of the biological replicates of each group was acceptable.

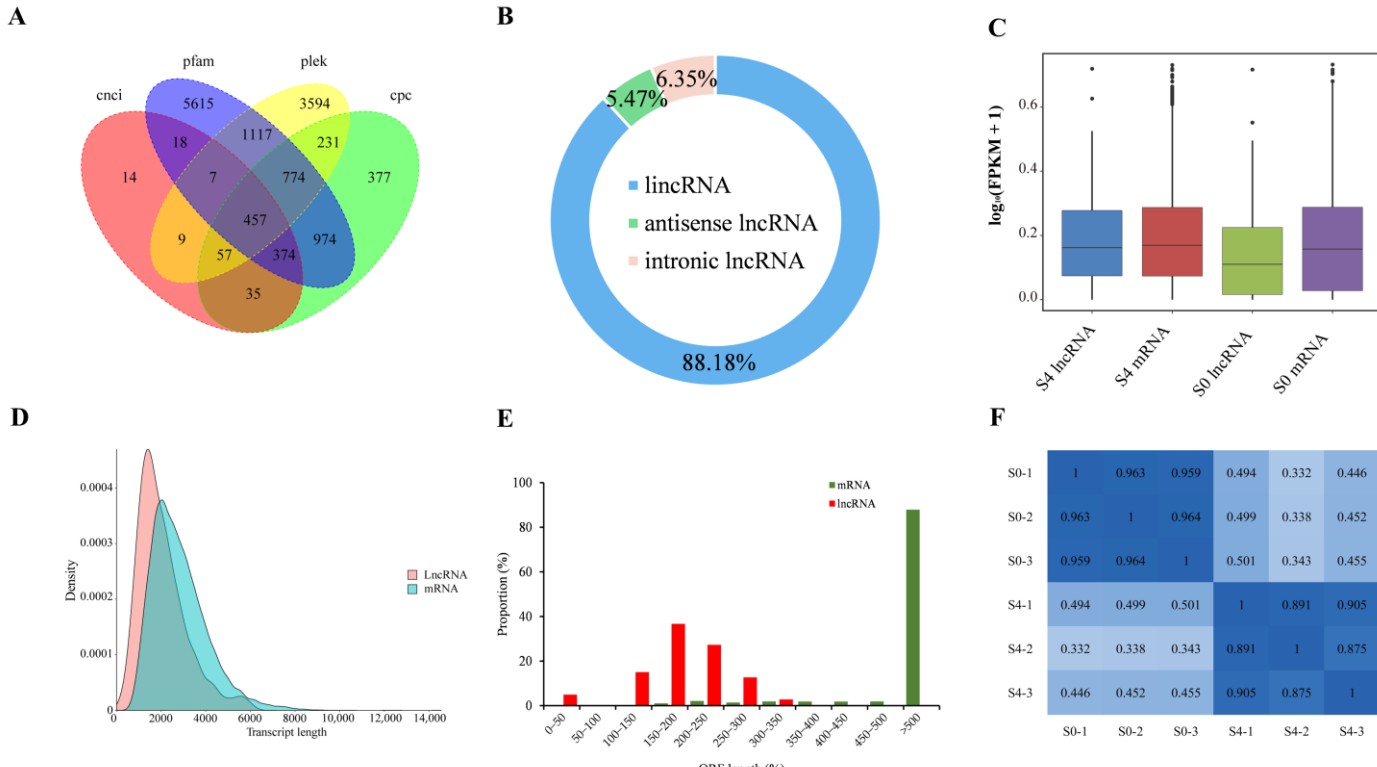

**Figure 1.** Identification and characterization of lncRNAs in M. glyptostroboides seeds under aging (S4) and normal treatment(S0). (**A**) Venn diagram showing the numbers of novel lncRNAs identified by coding-potential calculator (CPC), coding non-coding index (CNCI), the Protein Families database (Pfam) and Predictor for long non-coding RNAs and messenger RNAs based on an improved k-scheme (plek). (**B**) Classification of the lncRNAs identified in the study. (**C**) Overall expression levels Log10 (fragments-per-kilobase of transcript-per-million mapped reads (FPKM)+1) of lncRNAs and mRNAs in samples grown under the two treatments. (**D**) Distribution of lncRNAs and mRNA based on length. (**E**) Distribution of lncRNAs and mRNA based on ORF length. (**F**) Sample correlation coefficient clustering heat-map.

### 3.3. Analysis of Differentially Expressed Genes and LncRNAs

To determine the differential gene expression patterns of *M. glyptostroboides* seeds in response to aging stress, DESeq2 (v1.32.0) software was used for differential expression analysis by pairwise comparisons among the biological replicates per group. Under the conditions of $|\log2foldchange \geq 1|$ and $p$ value $< 0.05$, 9000 DEGs were identified in the analysis, in which S0 group was compared with S4 group. The expression of 1999 and 7001 genes were down-regulated and up-regulated, respectively (Figure 2A, Table S4). Moreover, 128 significantly differentially expressed lncRNAs (DELs) were observed, 94 up-regulated and 34 down-regulated (Figure 2B, Table S4).

The up- and down-regulated DEGs were subjected to GO function and KEGG pathway enrichment analysis to investigate the functions and biological pathways during the artificial aging of seeds. Enrichment analysis showed that 891 GO terms were involved in molecular functions (Figure 3A, Table S5), of which 67 were significantly enriched ($p$-value $< 0.05$) in this class. A total of 408 GO terms involved cellular components, of which 16 were significantly enriched ($p$-value $< 0.05$) in this class. Furthermore, 1789 GO terms were involved in biological process class, and among them were 201 GO terms including "nucleobase-containing compound biosynthetic process". GO:0034654, "transcription, DNA-templated" GO:0006351, "RNA biosynthetic process" GO:0032774, and "regulation of protein modification process" GO:0031399 were significantly enriched ($p$-value $< 0.05$). The enrichment of GO functions in the down-regulated DEGs showed that (Figure 3B, Table S6) 582 GO terms were involved in molecular functions, of which 93 were significantly enriched ($p$-value $< 0.05$)

in this class. A total of 278 GO terms involved a cellular component, of which 21 were significantly enriched ($p$-value < 0.05) in this class. A total of 1789 GO terms were involved in biological process class, of which 60 GO terms were significantly enriched ($p$-value < 0.05) in this class, including "mRNA methylation" GO:0080009, "methylation" GO:0032259 and "mRNA methyltransferase activity" GO:0008174. In general, seed deterioration is commonly accompanied by chromosome aberrations, telomere length changes, DNA damage, and DNA methylation. In this research, the differentially expressed gene GO term analysis indicated that these differentially expressed genes could be involved in repairing the damaged DNA to a certain extent and thus conferring the ability to germinate on seeds under abiotic stress [32]. KEGG Pathway enrichment analysis revealed that up-regulated DEGs are involved in 117 pathways, among which 12 pathways were significantly enriched ($p$-value < 0.05), including "Oxidative phosphorylation", "Protein processing in endoplasmic reticulum", "Citrate cycle (TCA cycle)", and multiple amino acid metabolism and degradation (Figure 3D, Table S7). In addition, some pathways related to seed aging were also found, such as "Pyruvate metabolism", "Pyruvate metabolism", "Galactose metabolism", "Glutathione metabolism", and "Peroxisome". In the down-regulated DEGs, KEGG pathways were found to be enriched in 92 pathways, and significantly enriched in 12 pathways ($p$-value < 0.05), such as "ABC transporters", "RNA polymerases", and "Linoleic acid metabolism" (Figure 3E, Table S8).

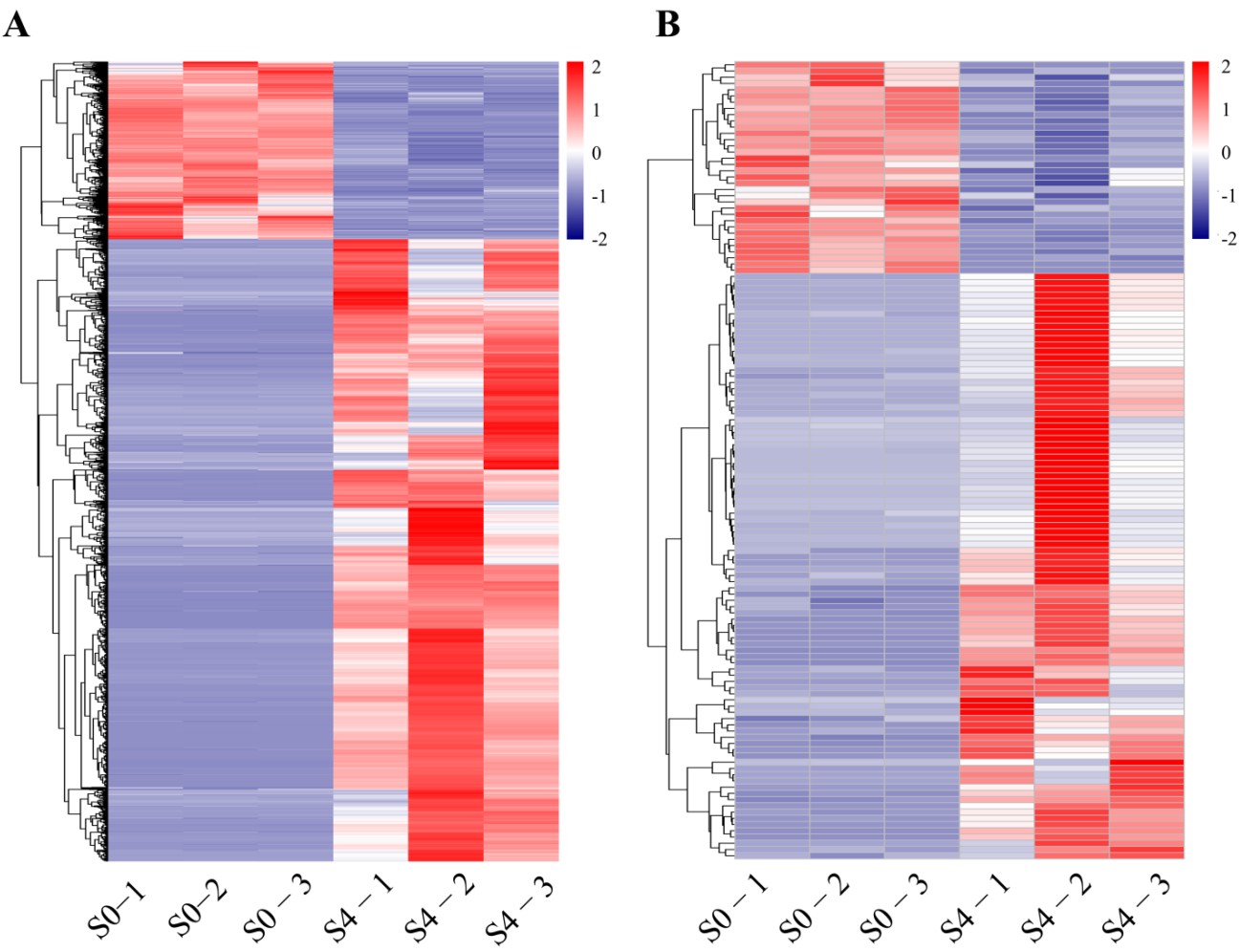

**Figure 2.** Transcript analysis of *M. glyptostroboides* seeds under S0 and S4, with three replicates. (**A**) Cluster analysis of specifically-expressed mRNAs. (**B**) Cluster analysis of specifically-expressed lncRNAs.

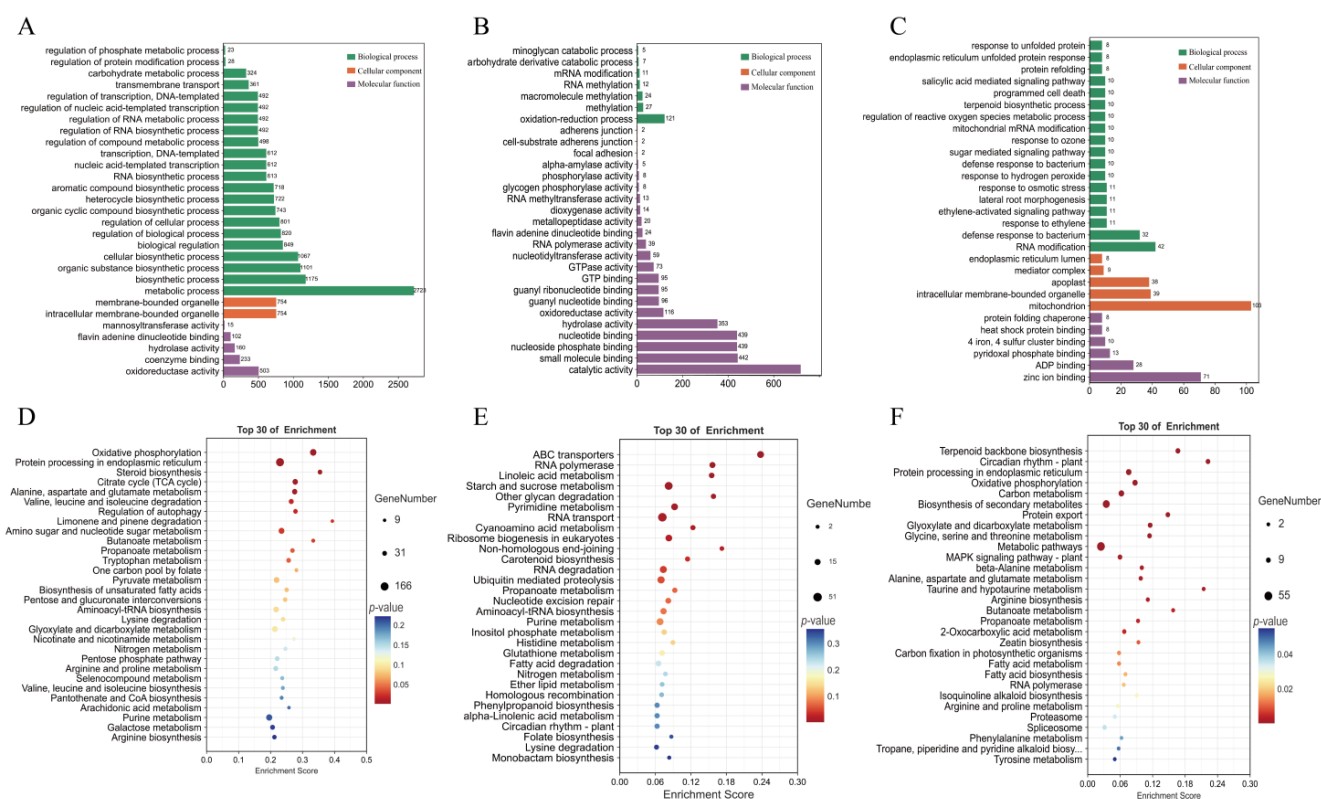

**Figure 3.** GO and KEGG analysis of the biological function of genes, LncRNAs, and target genes of LncRNAs. GO terms of up- (**A**) and down-regulated genes (**B**), (**C**) cis-target genes of lncRNAs, KEGG pathway enrichment analysis of genes, mRNA KEGG pathway of up- (**D**) and down-regulated mRNAs (**E**), and cis-regulated target genes of lncRNAs (**F**).

It has been shown that lncRNAs can function through cis- and trans-regulation, so exploring the functions of their target genes can predict the potential functions of lncRNAs. We predicted 844 cis-regulation genes of the 128 DELs using sequence alignment, and identified the functions of cis-regulation genes by GO enrichment analysis. GO enrichment analysis showed that cis-regulation genes involved 447 GO terms (Figure 3C, Supplementary Table S9), among which 51 GO terms were significantly enriched ($p$-value < 0.05) in the molecular functions class, including "protein folding chaperone" GO:0044183, "heat shock protein binding" GO:0031072, "DNA binding" GO:0003677, and binding function and enzyme activity. A total of 36 GO terms were significantly enriched ($p$-value < 0.05) in the cellular component class, involving multiple organelles, protease complexes, mRNA enzymes, and other terms. A total of 114 GO terms were significantly enriched ($p$-value < 0.05) in the biological process class, involving multiple stress-related biological processes, such as "RNA modification" GO:0009451, "defense response to bacterium" GO:0042742, "programmed cell death" GO:0012501, "regulation of reactive oxygen species metabolic process" GO:2000377, and "response to hydrogen peroxide". KEGG enrichment analysis showed that 55 pathways were enriched, of which 29 pathways were singificantly enriched, such as "Terpenoid backbone biosynthesis", "Protein processing in endoplasmic reticulum", "Oxidative phosphorylation", and "MAPK signaling pathway—plant" (Figure 3F, Table S10).

The expression levels of lncRNAs are correlated with the potential target gene, and the functions of lncRNAs can be predicted by the associated mRNAs in the lncRNA-mRNA co-expression network. The correlation between DELs and DEGs (correlation coefficient PCC $\geq$ 0.95, $p$-value < 0.01) were analyzed to identify the trans-regulated genes involved in the aging-stress response. A total of 8098 DEGs were reverse trans-regulated by 128 DELs, forming 108,215 lncRNA-mRNA gene pairs (Table S11).

The GO functions of these trans-regulated genes were enriched in 1248 GO terms (Figure 4A, Table S12), of which 353 GO terms were involved in molecular functions, and 69 GO terms related to enzyme catalysis and binding were significantly enriched (*p*-value < 0.05) in this class; 188 GO terms were involved in cellular component, of which 46 GO terms are significantly enriched (*p*-value < 0.05) in this class. A total of 705 GO terms referred to biological process class, of which 124 GO terms were significantly enriched (*p*-value < 0.05) in "protein folding" GO:0006457, "oxidation-reduction process" GO:0055114, "DNA repair" GO:0006281, "regulation of stomatal closure" GO:0090333, and biological processes in response to abiotic stresses. The KEGG functionally enriched 110 pathways (Figure 4B, Table S13), of which 43 pathways were significantly enriched (*p*-value < 0.05) in the metabolic pathways "protein processing in endoplasmic", "ubiquitin mediated proteolysis", etc. The results of GO-function enrichment and KEGG pathway enrichment of trans-regulated genes are similar to those of mRNA, suggesting that lncRNA can play an important role in these functions through trans-regulation in response to seed-aging stress.

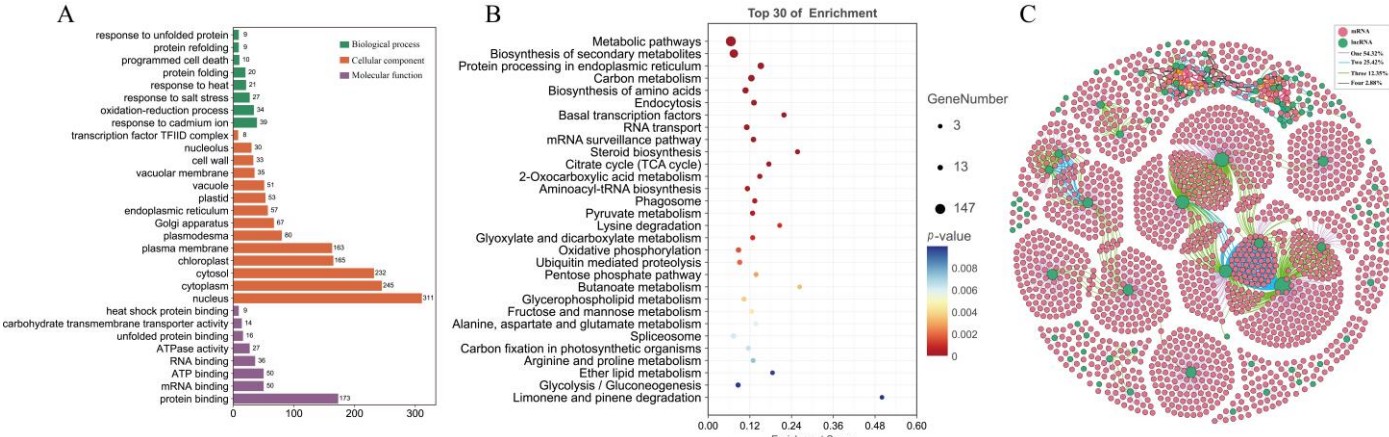

**Figure 4.** Interactions between DELs and their trans-regulated DEGs in *M. glyptostroboides.* (**A**) GO enrichment analysis of lncRNA trans-target genes. (**B**) KEGG pathway enrichment analysis of the trans-target gene. (**C**) Gephi network of differentially-expressed lncRNAs and co-expressed protein-coding genes: red circles represent lncRNAs and green circles represent protein-coding genes, with the size and complexity of the network reflecting the number of interactions involved.

In order to explain the function of lncRNAs and the relationship between lncRNAs and protein-coding genes during seed aging of *M. glyptostroboides* (Figure 4C), Gephi was further used to establish an interaction network between annotated co-expressed protein coding genes and DELs. The analysis revealed a very complex relationship between protein-coding genes and lncRNAs. The analysis showed that there was a very complex relationship between protein-coding genes and lncRNAs. One lncRNA can interact with multiple protein-coding genes, and one protein-coding gene can also interact with multiple lncRNAs. Obviously, lncRNAs interacting with multiple protein-coding genes may play an important role in the seed-senescence-stress response. The lncRNAs that interact with multiple protein-coding genes may play an important role in the seed-aging stress response.

### 3.4. Potential lncRNA-miRNA-mRNA ceRNA Network in M. glyptostroboides Seeds under Aging Stress

In this study, the lncRNAs and mRNAs of the co-expressed network were used as prediction libraries of ceRNA and mRNA targets of miRNAs. The psRNATarget server was used to identify DELs as the potential targets of miRNAs. First, it was predicted that 47 miRNAs could be decoyed by 23 lncRNAs, forming 59 lncRNAs-miRNA pairs. A total of 305 miRNAs were then predicted to target 750 mRNAs, forming 1060 miRNA-mRNA pairs. Finally, the above two related data sets were cross-referenced, and the lncRNAs and mRNA pairs that shared the same miRNA were selected to construct the ceRNA network.

The entire ceRNA network contained 38 miRNAs, 18 lncRNAs, and 69 mRNAs (Figure 5, Table S14), of which the interconnected network with the most members were visualized using Cytoscape. Some miRNAs and mRNAs related to seed viability and abiotic stress are also in this network, such as the *miR164 family, miR167 family, miR168, miR399, PKL, UKL, HXK2, BIP3,* and *RCD1* (Table S15) [34–37]. KEGG pathways showed that mRNAs in the ceRNA network were mainly enriched in alanine, aspartate, and glutamate metabolism, protein processing in the endoplasmic reticulum, galactose metabolism, etc. (Table S16). GO enrichment analysis of miRNA-related mRNAs showed that the miRNAs were mainly involved in 117 biological processes, such as "DNA methylation-dependent heterochromatin assembly" GO:0006346, "response to cadmium ion" GO:0046686, "cellular glucose homeostasis" GO:0001678, and "positive regulation of protein catabolic process" GO:0045732 (Table S17). It is worth noting that the main enriched genes in the GO term "programmed cell death" GO:0012501 were the oxidative-stress-response gene *RCD1* (Radical-Induced Cell Death 1), an important transcriptional regulator (namely, mgl-mRNA039425, mgl-mRNA010409, mgl-mRNA024224, mgl-mRNA031164, mgl-mRNA036477, mgl-mRNA026561). These genes and the miRNA167 family, lncRNA_00185, constitute a ceRNA network, which may play an important role in seed aging.

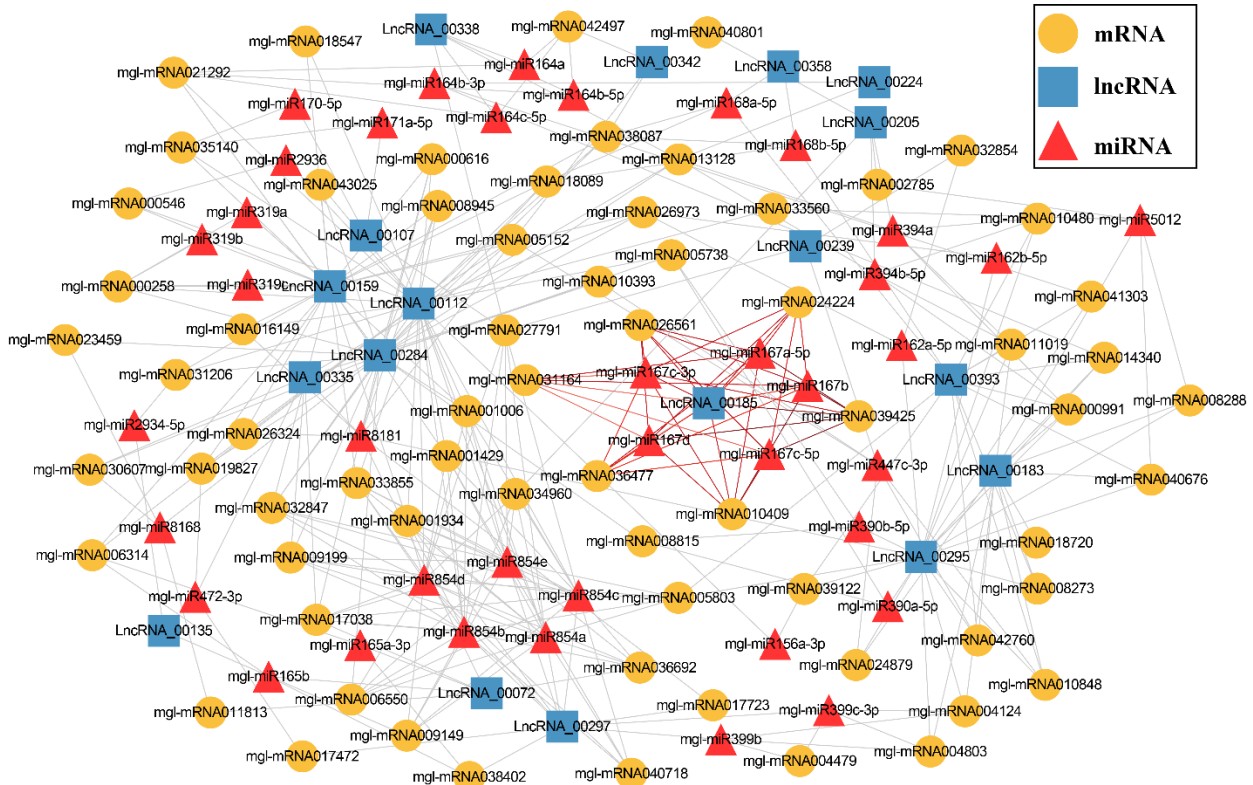

**Figure 5.** CeRNA networks among lncRNAs, mRNAs, and miRNAs. The red triangles represent miRNAs, the yellow circles represent mRNAs, and the peacock-blue square represent lncRNAs.

### 3.5. Validation of Differentially-Expressed lncRNAs by qPCR

To verify the reliability of the sequencing results, we randomly selected six DELs (LncRNA_00123, LncRNA_00236, LncRNA_00102, LncRNA_00272, LncRNA_00119, LncRNA_00189) for validation (Figure 6, Table S18). The results showed that, although the up- or down-regulation of six DELs was different from the sequencing results, the trend of gene expression was consistent with the RNA-Seq results.

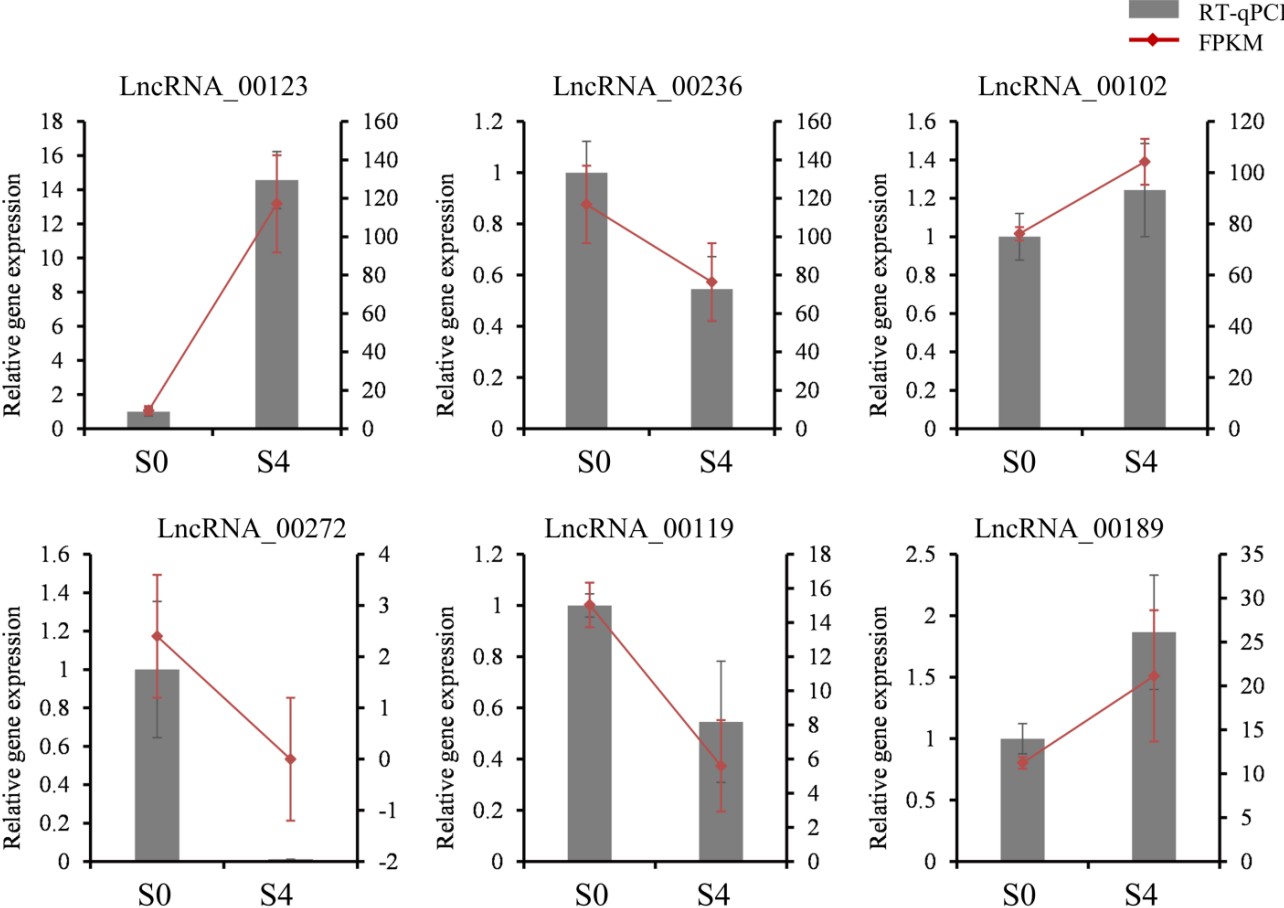

**Figure 6.** Relative expression analysis of select lncRNAs using quantitative real time polymerase reaction.

## 4. Discussion

### 4.1. Full-Length Sequences Identified by SMRT Sequencing in M. glyptostroboides Provided Resources for Studies of the Aging Stress Response

By the end of 2020, it has been reported that genome-wide fine maps and sketches were available for only 843 plant species [38]. Due to rapid advances in sequencing technology, the speed and quality of sequencing continue to improve and the cost of sequencing has been significantly reduced. Nevertheless, the whole genome assembly and sequencing of large genome plants such as *Sequoia sempervirens*, *M. Glyptostroboides*, and *S. Giganteum* are still a huge challenge. The lack of high-quality genomic maps limits the research of plants with large genomes [30]. The full-length transcriptome sequencing technology (ISO Seq, isoform sequencing), that is, single-molecule real-time sequencing technology, has the advantages of an ultra-long read length, no template amplification, a short running time, the direct detection of apparent modification sites, and the direct detection of transcripts [31]. The use of SMRT sequencing technology can provide an ultra-long reference sequence for RNA-Seq research without a reference genome, greatly reducing the number of transcripts generated by the traditional second-generation RNA Seq without reference genome, narrowing the scope of research, promoting the research development in the plant field, and providing a new possibility for the research of some plants without sequenced genomes [31]. In this study, PacBio's SMRT was used to obtain full-length transcript information from mixed samples of *M. glyptostroboides*. We then combined SMRT sequencing with Illumina sequencing. We corrected the long reads of SMRT with short reads and obtained 42,189 complete transcripts of *M. glyptostroboides*. Subsequently, we compared all transcripts with NR, NT, Swiss prot, GO, COG, KOG, Pfam, and KEGG and found that, out of the transcripts obtained, a total of 40,446 tran-

scripts (95.87%) were successfully annotated, indicating that SMRT sequencing greatly improved the accuracy and depth of transcriptome research. At present, since the genome of *M. glyptostroboides* has not been published yet, we selected its closest relative *S. giganteum* in the NR database as a reference and divided it into four categories by computational analysis to explore its possible functions. Moreover, we also found that the average length of lncRNAs is longer than that of mRNAs, which is consistent with previous studies [39]. This is also the first report of using SMRT to study lncRNAs in response to *M. glyptostroboides* seed aging. The full-length transcriptome sequence obtained will facilitate research on the genetics and physiology of *M. glyptostroboides*.

### 4.2. LncRNAs Regulate Genes in Response to Aging Stress in M. glyptostroboides Seeds

Long non-coding RNAs play important roles in many biological processes in plants, such as root development, light response at the seedling stage, flowering time control, reproductive development, stress response, and disease resistance [23–25,40–42]. In this study, we identified 457 putative *M. glyptostroboides lncRNAs* in S0 and S4 samples by full-length transcriptome sequencing. There is evidence that, in eukaryotes, lncRNAs can function in a cis or trans manner, affecting the structure and function of adjacent chromatin or chromatin-related proteins, bringing enhancers and promoters to their targets, and jointly regulating the expression network [43]. By using sequence complementation and trans regulation, we predicted 844 cis-target genes and 8098 trans-target genes, and performed GO enrichment analysis. It was found that the defense response, regulation of reactive oxygen species metabolic process (GO: 2,000,377), and programmed cell death (GO: 0,012,501) were specifically enriched. In addition, KEGG is also significantly enriched in metabolic pathways such as "Oxidative phosphorylation", "Protein processing in endoplasmic reticulum", and "MAPK signaling pathway—plant". This result is not only similar to the enrichment results of up- and down-regulated mRNA in KEGG and GO functions, but also consistent with the results of other crop-seed aging research [44–49].

Previous studies have suggested that the production and clearance of reactive oxygen species (ROS) in normal cells are in a state of dynamic equilibrium. During the seed-aging process, the main characteristic is that the production rate of ROS is greater than the clearance rate, which leads to the accumulation of intracellular ROS, and, subsequently, a series of physiological and biochemical reactions, such as nucleic acid degradation, lipid membrane degradation, and protein degradation, can occur [44,45]. The antioxidant enzymatic system in plants is of great significance for eliminating intracellular reactive oxygen species and delaying seed senescence [48]. In this study, it was found that the up-regulated genes were enriched in the ascorbic acid glutathione cycle, peroxisome, and other antioxidant pathways, indicating that during the process of seed aging, the activity or content of antioxidant enzymes decreased in the process of eliminating the oxidation effect of ROS and a large amount of expression was required to maintain cellular homeostasis. It has been shown that oxidative stress caused by seed aging leads to the accumulation of protein misfolding and the dysfunction of organelles, which in turn accelerates programmed cell death [50]. We found that the GO term "response to unfolded protein" was significantly enriched in both cis-target genes and trans-target genes, among which the expression of *Bip3* gene (endoplasmic reticulum-bound chaperone luminal-binding protein 3, *BIP3*) was significantly up-regulated.Some studies showed that the normal protein-folding of the endoplasmic reticulum is impaired during the aging process. The study also found that *BiP* synthesis is enhanced if protein folding in the endoplasmic reticulum is disturbed. *BiP* can prevent polymer formation by binding to misfolded proteins and facilitate correct refolding [51–54]. Chen et al. found that it can cause endoplasmic reticulum stress in pea-seed cells after artificial seed aging, and its marker protein is *BiP* protein [55]. Cao et al. also found that artificial aging treatment can cause endoplasmic reticulum stress in maize-seed embryo cells [56]. The folding and refolding of proteins in the endoplasmic reticulum is a highly energy-intensive process. The misfolding of proteins consumes large amounts of ATP and may stimulate mitochondrial oxidative phosphorylation, increasing ATP and ROS

production [57]. The ROS produced by mitochondria may in turn strengthen the response to ER stress, thereby enhancing the accumulation of mitochondrial ROS, which is also a potential signaling mechanism for the related ROS produced by endoplasmic reticulum (ER) stress and mitochondrial dysfunction [58]. Therefore, according to the results of this study, it is speculated that during the aging process of *M. glyptostroboides* seeds, the production and accumulation of ROS causes endoplasmic reticulum stress, resulting in the incorrect formation of disulfide bonds and the production of unfolded and misfolded proteins. In order to repair these products, the endoplasmic reticulum may generate a large amount of ROS in the process of promoting the formation of correct disulfide bonds in unfolded and misfolded proteins. Endoplasmic reticulum stress signals act on mitochondria, further aggravating the production of mitochondrial ROS, leading to an increase in the level of ROS in cells, which in turn causes severe damage to cells, and lncRNAs are involved through cis- and trans-regulation. There is also galactose metabolism related *HKX2* genes and other genes in the programmed cell death pathway. It can be speculated that the surge of ROS level leads to the up-regulation of genes in the galactose metabolism pathway, fat anabolism, and the amino acid degradation pathway, and ultimately aggravates the aging of *M. glyptostroboides* seeds.

In general, the GO term or KEGG pathways involved in the homeopathic regulation and trans-regulation of target genes by lncRNAs are very close to the pathways involved in the enrichment of mRNA KEGG, and these pathways are also consistent with previous studies on seed aging [44–49]. This indicates that the function of lncRNAs during the aging process of M. glyptostroboides seeds may be to respond to the aging process of seeds by regulating the genes of these pathways. Next, specific lncRNAs will be screened as indicators of the *M. glyptostroboides* seed-aging response.

*4.3. ceRNA Network Analysis in Response to M. glyptostroboides Seed under Aging Stress*

In addition to directly participating in the regulation of gene expression, another important function of lncRNA is to participate in the regulation of gene expression by interacting with miRNA response elements (MRes) and forming a competitive endogenous RNA (ceRNA) regulatory network with other RNA molecules [59]. MicroRNAs (miRNAs) are ubiquitous in eukaryotic genomes. They are a class of small noncoding RNAs composed of 20–24 nucleotides, which usually degrade, or inhibit the translation process of, their target genes after transcription by sequence complementation, thereby regulating the expression of genes at the transcriptional and post-transcriptional levels [60]. A large number of studies have shown that plant miRNAs play an important role in regulating plant growth and development and responding to stress [59,61–63]. However, studies on the regulatory relevance of miRNAs involved in seed vigor or anti-aging ability are rare. At present, only miR156, miR172, miR402 in Arabidopsis, and miR398b in tobacco have been found to be associated with seed vigor. In rice, the expression of miR164c was negatively correlated with seed vigor, while the expression of miR168a was positively correlated with seed vigor [35,64,65]. Other studies have shown that the over-expression of miR406 leads to flowering delay and seed-germination failure [66,67].

At present, the function of lncRNA as miRNA response element (MRes) has been well studied. For example, Yang et al. constructed the ceRNA network of Shanlan rice in response to drought. *Mstrg.28732.3* constituted the ceRNA network by targeting the common *miR171*. Functional analysis showed that they could play a role in the synthesis of the chlorophyll membrane in plants, thus affecting the ability of plants to resist drought stress [68]. In tomato, the *lncRNA 23468-miR482b-nbs-lrr* network can affect plant disease resistance, and the expression level of *miR482b*, as a negative regulatory factor, is reduced after inoculation of Phytophthora infestans in tomato [69]. In winter wheat, *lncR9A*, *lncR117*, and *lncR616* competitively bait *miR398* and regulate the expression of csd1 (the target gene of *miR398*) to affect cold tolerance [70]. In addition, the ceRNA network has also been found in Melilotus albus under salt stress and Arabidopsis under blue-light-induced stress [71]. In this study, a ceRNA network of 38 miRNAs, 18 lncRNAs, and 69 mRNAs was constructed,

and some miRNAs and mRNAs associated with seed viability and abiotic stress were found to be present in this network, such as the *miR164 family, miR167 family, miR168, miR399, PKL, UKL, HXK2,* and *BIP3*. Notably, the mRNAs involved in programmed cell death in GO enrichment were mainly *RCD1*, which was associated with the "RNA degradation" pathway (mgl-mRNA039425, mgl-mRNA010409, mgl-mRNA024224, mgl-mRNA031164, mgl-mRNA036477, mgl-mRNA026561) genes, and the ceRNA network was *miR167(a, b, c, d)-lncRNA_00185-RCD1*.

It has been shown that *miR167* plays an important regulatory role in the morphogenesis, growth and development, and hormone secretion of plant-seed embryos [72–74]. Notably, *miR167* was identified to be characteristically enriched during plant senescence several times by previous authors [75,76]. Cheng et al. found *miRNA167s* were up-regulated in rice seeds after 14 days of aging [77], and several studies have shown that the *RCD1* gene plays a role in biological stress, growth and development, and plant-senescence processes by positively regulating ROS delivery and binding to multiple transcription factors [78–81]. It has been shown that the RCD1 gene plays an important role in seed longevity; for example, FAIZA ALI found that *RCD1*-related loss of function Arabidopsis mutants *rcdl-3, srol-1, srol-2*, and *dreb2b* had significantly enhanced seed viability and longevity [82]. In this study, the expression of lncRNA_00185 and *RCD1* (mgl-mRNA039425, mgl-mRNA010409, mgl-mRNA024224, mgl-mRNA031164, mgl-mRNA036477, mgl-mRNA026561) were both significantly down-regulated; therefore, it is speculated that *miR167(a,b,c,d)* may compete for the lncRNA_00185 binding site to regulate the expression of the *RCD1* genes, thereby maintaining the homeostasis of seed reactive oxygen species to improve seed-aging resistance. However, further experiments are needed to uncover the specific biological processes involved.

## 5. Conclusions

In this study, the full-length transcriptome sequence of *M. glyptostroboides* was obtained by SMRT for the first time, the lncRNAs during the artificial aging of *M. glyptostroboides* seeds were identified, and its basic characteristics and potential functions were extensively studied. To our knowledge, this is the first report on aging-responsive lncRNAs in *M. glyptostroboides* seeds. These findings will provide a useful basis for in-depth functional characterization of lncRNAs in *M. glyptostroboides* seeds under aging stress. In this study, it was identified that some key pathways involved in the aging process of *M. glyptostroboides* seeds are regulated by lncRNAs, among which are protein processing in the endoplasmic reticulum pathway, the programmed cell death GO term are enriched in mRNA, cis-regulatory target genes, trans-regulatory target genes, and the ceRNA network, indicating that endoplasmic reticulum stress of *M. glyptostroboides* embryo cells may be the main factor causing the decrease in seed vigor during artificial aging treatment. This study also constructed a ceRNA network, in which *miR167(a,b,c,d)*-lncRNA_00185-*RCD1*(mgl-mRNA039425, mgl-mRNA010409, mgl-mRNA024224, mgl-mRNA031164, mgl-mRNA036477, and mgl-mRNA026561 were key candidates that may be involved in response to reactive oxygen species homeostasis during seed aging through ceRNA-network interaction, which is worthy of further research in the future.

**Supplementary Materials:** The following supporting information can be downloaded at: https://www.mdpi.com/article/10.3390/f13101579/s1, Figure S1: *M. glyptostroboides* long-read sequencing transcriptome annotation with different databases; Figure S2: Homologous species distribution of *M. glyptostroboides* annotated in the NR database; Table S1: Primers used for q-PCR analysis; Table S2: Statistics of 6 samples data filtering of RNA-Seq; Table S3: All mRNA and nr, SwissProt, KEGG, KOG, GO, NT, and pfam database annotation results; Table S4: Significantly DEGs and DELs during seed aging; Table S5: Enrichment analysis of Gene Ontology analysis for all up-regulated mRNA; Table S6: Enrichment analysis of Gene Ontology analysis for all down-regulated mRNA. Table S7: Kyoto Encyclopedia of Genes and Genomes (KEGG) pathway analysis for all up-regulated mRNA; Table S8: Kyoto Encyclopedia of Genes and Genomes (KEGG) pathway analysis for all down-regulated mRNA; Table S9: Enrichment analysis of Gene Ontology analysis for all cis-target mRNA; Table S10:

Kyoto Encyclopedia of Genes and Genomes (KEGG) pathway analysis for all target genes of DELs in *M. glyptostroboides*; Table S11: Correlation analysis between lncRNA and mRNA. Table S12: Enrichment analysis of Gene Ontology analysis for all trans-target mRNA; Table S13: Kyoto Encyclopedia of Genes and Genomes (KEGG) pathway analysis for all target genes of DELs in *M. glyptostroboides*; Table S14: The interacted pairs of lncRNAs, miRNAs, and target mRNAs in ceRNA network. Table S15: MiRNA-mediated annotation of mRNAs interacting with lncRNAs. Table S16: GO enrichment analysis of mRNA having miRNA mediated interactions with lncRNA. Table S17: KEGG enrichment analysis of mRNA having miRNA mediated interactions with lncRNA; Table S18: Quantitative reverse transcription PCR raw data.

**Author Contributions:** Y.L. participated in performing the experiments, data analysis, drafting and revising the manuscript; J.L. (Jingyu Le) participated in performing the experiments, data analysis, drafting the manuscript; Y.Z. par-ticipated in drafting and revised the manuscript; R.W. participated in performing the experiments; Q.L. participated in drafting the manuscript; X.L. participated in drafting the manuscript. J.L. (Jun Liu) participated in drafting and revising the manuscript. Z.D. participated in conceiving the study and revising the manuscript. All authors have read and agreed to the published version of the manuscript.

**Funding:** This work was supported the National Natural Science Foundation of China (31860073, 31871716), the Science and Technology Program of Guangdong Province, China (2020B121201008, 2019KJ106), the Science and Technology Program of Guangzhou (201909020001) and the Foundation of Guangdong Academy of Agricultural Sciences (202132TD).

**Data Availability Statement:** Transcriptome datasets supporting the conclusions of this article are available in the NCBI BioProject repository under the accession number PRJNA872838.

**Acknowledgments:** Thanks to Zeng Jianming from the University of Macau and his team for their technical guidance. Thanks for the sequencing service provided by NovoGene. Thanks to Yongpeng Luo, Zhangyan Dai, and Yuhai Cui for revising the manuscript.

**Conflicts of Interest:** The authors declare no conflict of interest.

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
