# Peer review of "Identification and Functional Analysis of LncRNAs in Response to Seed Aging in Metasequoia glyptostroboides by Third Generation Sequencing Technology"

_forests, doi:10.3390/f13101579_

Round 1

Reviewer 1 Report

This manuscript analyzing lncRNA changes during Metasequia glyptostroboides seed aging. The results seem to be interesting based on data analysis. But, I think some aspects still need to be revised.

1  Fist, there are many mistakes in the formatting of this manuscript: “Making it difficult (Line 61), “Boththe aged (Line 118)”, “seeds of M. glyptostroboidesvulgaris (Line 131)”, “sizes of 1-4kb and >4kb to library (Line 131)”, “by 2-DDCt (Line 222)” and the last word of each line is poorly formatted from Line 154 to 167. Please carefully check the from of words or sentences throughout the manuscript.

2  In Introduction section, some references were cited by the authors as the formats of “found that” in Line86, Line 88, Line 93, Line 95. I strongly recommended that the authors cite the references in a different way.

3  What do these numbers mean in Line 238? “The RNA-seq generated 119930400 and 125220590 raw data from the S0 and S4 samples”

4  I am confused by the sentence of “A total of 8098 DEGs were trans-regulated with 128 DELs, of which 108215 lncRNA-mRNA gene pairs were positively correlated and 76048 lncRNA-mRNA gene pairs were negatively correlated”. What do the authors mean?

5  Please carefully checked the sentence of “the trend of gene expression the trend of gene expression was were consistent with the RNA-Seq results, which and werewas in accordance with the characteristics of seed aging period. (Line 435 to 437)”. I am very confused, what is the author trying to express?

6  I suggested that some sentences from Line 444 to Line 459 in the discussion be moved to the introduction.

7  There are errors in the language that could be improved by a native English speaker.

Author Response

This manuscript analyzing lncRNA changes during Metasequia glyptostroboides seed aging. The results seem to be interesting based on data analysis. But, I think some aspects still need to be revised.

Response:We are very grateful to your comments on the manuscript. All of your questions and comments have been addressed (see below).

  • First, there are many mistakes in the formatting of this manuscript: “Making it difficult (Line 61)”, “Boththe aged (Line 118)”, “seeds of M. glyptostroboidesvulgaris (Line 131)”, “sizes of 1-4kb and >4kb to library (Line 131)”, “by 2-DDCt (Line 222)” and the last word of each line is poorly formatted from Line 154 to 167. Please carefully check the form fo of words or sentences throughout the manuscript.

Response:We have revised our manuscript accordingly. The manuscript has also been double-checked, and the typos and grammar errors we found have been corrected.

  • In Introduction section, some references were cited by the authors as the formats of “found that” in Line86, Line 88, Line 93, Line 95. I strongly recommended that the authors cite the references in a different way.

Response: We have now worked on both language and readability and have also involved native English speakers for language corrections. We believed the language level have been substantially improved.

  • What do these numbers mean in Line 238? “The RNA-seq generated 119930400 and 125220590 raw data from the S0 and S4 samples”

Response: What we intended to say is that “The RNA-seq experiments generated approximately 119930400 and 125220590 new reads from the S0 and S4 samples, respectively”. We have revised in the text (see line 243-244)

  • I am confused by the sentence of “A total of 8098 DEGs were trans-regulated with 128 DELs, of which 108215 lncRNA-mRNA gene pairs were positively correlated and 76048 lncRNA-mRNA gene pairs were negatively correlated”. What do the authors mean?

Response: Thank you for pointing this out, we have corrected this error: in lines365-366,we removed“and 76048 lncRNA-mRNA gene pairs were negatively correlated”, and added the data related to the lncRNA and mRNA to Supplementary Table S11。

  • Please carefully checked the sentence of “the trend of gene expression the trend of gene expression was were consistent with the RNA-Seq results, which and werewas in accordance with the characteristics of seed aging period. (Line 435 to 437)”. I am very confused, what is the author trying to express?

Response: We have carefully checked the results, modified this sentence to “the trend of gene expression was consistent with the RNA-Seq results”.

6 I suggested that some sentences from Line 444 to Line 459 in the discussion be moved to the introduction.

Response: We gratefully appreciate your valuable suggestion. As suggested, we have moved the sentences from Line 444 to Line 459 in the discussion to the Introduction. See line 103-108

7  There are errors in the language that could be improved by a native English speaker.

Response: Thanks a lot your comments. The manuscript has been revised by a native English speaker.

Reviewer 2 Report

This manuscript uses single-molecule real-time (SMRT) sequencing technology combined with illumina RNA-seq to analyze changes in lncRNAs during Metasequoia seed aging. These results have important reference value for elucidating the molecular mechanism of Metasequoia seed aging, improving the storage ability of crop seeds, and protecting rare germplasm resources. This paper provides detailed omics data for the research field. I suggest that it will be accepted after the following minor revisions.

1.     2.2. Extraction of total RNA describes RNA extraction. But the manuscript should add the RNA concentration and integrity information in the Results.

2.     The study verified the accuracy of RNA-SEQ sequencing results using the qPCR platform, However, it does not provide the reference gene information and the raw data.

3.     Authors should analyze the functions of some lncRNAs by molecular technology.

Author Response

This manuscript uses single-molecule real-time (SMRT) sequencing technology combined with illumina RNA-seq to analyze changes in lncRNAs during Metasequoia seed aging. These results have important reference value for elucidating the molecular mechanism of Metasequoia seed aging, improving the storage ability of crop seeds, and protecting rare germplasm resources. This paper provides detailed omics data for the research field. I suggest that it will be accepted after the following minor revisions.

Response:Thank you for your positiveand constructive comments on our manuscript. We have carefully considered the suggestions and made changes accordingly.

Point 1: “2.2. Extraction of total RNA” describes RNA extraction. But the manuscript should add the RNA concentration and integrity information in the Results.

Response:We have made the suggested corrections according to the Reviewer’s comments. We have added the RNA concentration and integrity information in the Results. See line131-134.

Point 2: The study verified the accuracy of RNA-SEQ sequencing results using the qPCR platform, However, it does not provide the reference gene information and the raw data.

Response:The reference genes of q-PCR and lncRNA sequences were included in supplementary table S1. The raw data of q-PCR were in supplementary table S18. Expression data were in supplementary Table S4.

Point 3: Authors should analyze the functions of some lncRNAs by molecular technology.

Response:Thank you for the great suggestions. In this manuscript, we focus on analyzing lncRNA changes during Metasequia glyptostroboides seed aging. Next, we will investigate the biological functions of the lncRNAs of special interest. We plan to use various molecular approaches to examine the detailed functions and mechanisms and will be presented in our future publications.
